# Genome-Wide DNA Methylation and Transcription Analysis Reveal the Potential Epigenetic Mechanism of Heat–Light Stress Response in the Green Macro Algae *Ulva prolifera*

**DOI:** 10.3390/ijms26136169

**Published:** 2025-06-26

**Authors:** Kifat Jahan, Sylvia Kristyanto, Keun-Hyung Choi

**Affiliations:** Department of Earth, Environmental and Space Sciences, Chungnam National University, 99 Daehak-ro, Yusung-gu, Daejeon 34134, Republic of Korea; kifatjahan170@gmail.com (K.J.);

**Keywords:** *Ulva prolifera*, DNA methylation, transcriptomic, physiological changes, glycolysis pathway

## Abstract

*Ulva prolifera* (Chlorophyta), a pivotal species in green tide generation, is particularly vulnerable to abiotic stressors, including variations in temperature and light intensity, requiring specific regulatory frameworks for survival. Epigenetic modification is recognized as a molecular mechanism contributing to the flexible adaptability to environmental alterations. In this study, using DNA methylation pattern analysis, we investigated abiotic stress responsive methylation events, as well as gene and pathway expression patterns, in green macroalgae *U. prolifera* cultured under elevated temperature–light stress (30 °C and 300 µmol photons m^−2^ s^−1)^ and identified a negative correlation between CG methylation and gene expression patterns which indicated that abiotic stress caused CG demethylation and afterwards provoked the transcription response. CHG and CHH methylation exhibited an increased mutability and were preeminently found in transposable elements and intergenic regions, possibly contributing to genetic stability by restricting transposon activity. Furthermore, a rapid regeneration through spore ejection and the formation of new thalli was observed, which emphasized its tenacity capacity for stress memory. Our study also revealed an upregulation of genes associated with the glycolysis pathway and highlighted the critical roles of hexokinase, 6-phosphofructokinase-1, and fructose-6-phosphate in triggering glycolysis as a significant stress-adaptive pathway. Overall, these findings suggested that DNA methylation functions as a potential regulatory mechanism, maintaining environmental adaptability, genomic integrity, and underpinning regenerative capacity in *U. prolifera*. The findings elucidated the molecular resilience of *U. prolifera*, highlighting its feasibility for sustainable development and biotechnological applications.

## 1. Introduction

Anthropogenic global change, marked by persistent warming and heightened seasonal fluctuations, generates stressful conditions in shallow waters, presenting a considerable challenge for seaweeds [1,2,3]. A significant obstacle still exists in predicting how the biodiversity and evolution of aquatic creatures will be impacted by shifting ambient temperatures [4]. Environmental variables, including temperature, light, and salinity, will invariably influence the growth and reproduction of macroalgae [5,6]. Elevated temperatures and light intensity are the principal factors that limit the productivity of macroalgae in their habitat. In addition to the risk of physical injury, it can also influence numerous biological processes, including hormone levels, gene expression, protein synthesis, photosynthesis, respiration, water balance, and membrane stability [5,7]. Therefore, water temperature and light intensity are two of the most important abiotic factors that affect all life activities of aquatic animals [8,9,10]. 

Due to daily and seasonal fluctuations in temperature and light intensity, marine organisms frequently sustain injuries. Seaweed can sense both ephemeral and seasonal changes in the surrounding environment and can develop specialized defense mechanisms to safeguard against such threats. Some macroalgal species have demonstrated a remarkable ability to regenerate. Several studies have observed that in young thalli, the adjacent cell is capable of undergoing budding, rhizoid development, or regeneration after being damaged through biotic or abiotic stress [11,12,13,14,15]. At elevated temperatures and light intensity, the dimensions of algal cells, together with their protein and carbohydrate content, diminish [16]. To alleviate damage from various stresses, seaweeds depend on a sophisticated system of antioxidant enzymes and signaling molecules [17]. Increased temperature–light intensity can enhance the production of reactive oxygen species (ROS, H_2_O_2_), thereby causing significant oxidative damage to biological components, including proteins, DNA, and lipids [18]. In that scenario, CAT and SOD play significant roles. Through the effective management of reactive oxygen species (ROS) and the maintenance of energy equilibrium, seaweeds bolster their resilience to adverse abiotic circumstances, thereby assuring survival and growth in demanding marine settings.

*U. prolifera*, O.F. Müller, 1778, (Chlorophyta) [19] serves as a key species contributing to the phenomenon of green tides and holds considerable economic importance as a seaweed species [5,20,21]. This group is persistently subjected to various abiotic stresses that may adversely affect their growth, development, productivity, and survival [21,22]. This organism flourishes in a temperature range of 10–20 °C [18] and requires a light intensity between 60 and 140 μmol·m^−2^·s^−1^ [23]. The molecular response of *U. prolifera* to short-term high light stress was investigated using a multi-omics approach. Nevertheless, its development is constrained when the temperature falls below 10 °C or exceeds 20 °C [24,25,26] and when light intensity is below 40 μmol·m^−2^·s^−1^ or above 200 μmol·m^−2^·s^−1^ [15,27].

Researchers have conducted thorough investigations on *U. prolifera*, recognizing it as an important ecological indicator for assessing the effects of temperature and light intensity on multiple facets of its development, physiology, and biochemistry [23,28,29,30,31]. Numerous recent studies have examined the physiological responses of different marine macroalgae to abiotic stress, including *Bangia* sp. [32,33,34,35] and *Dictyota dichotoma* [14], *U. prolifera* [16,18,36]. Previous studies have indicated that the alterations in DNA methylation patterns facilitate swift and reversible adjustments in chromatin architecture, thereby promoting the activation of defense pathways via the interplay of genetic and epigenetic processes [37,38,39,40,41,42,43,44,45].

*U. prolifera* can acclimate to environmental stressors and changes via morphological and physiological adjustments, which contribute to its success in changing environments. Nevertheless, research concerning DNA methylation modification and epigenetic mechanisms has been rarely explored in *U. prolifera*. Nonetheless, the regeneration process of marine macroalgae *U. prolifera* under stressful conditions has been inadequately documented at both physiological and molecular levels, resulting in a limited understanding of the underlying mechanisms. This research focused on investigating how DNA methylation influences tolerance to abiotic stress, with the goal of elucidating the epigenetic regulatory mechanisms that are inextricably linked with the regeneration of green macroalgae (*U. prolifera*) under climate change conditions. This study employed methylation profiling in conjunction with assessments of physiological performance. This study aimed to elucidate the molecular mechanisms and examine the physiological changes essential for survival under conditions of temperature–light stress.

## 2. Results

### 2.1. Physiological Change

All treatment groups exhibited rapid regeneration within 24 h of stress (Figure 1A), and alterations in regeneration structure were noted throughout the entire study duration. Following a 3-day incubation of the explants at 30 °C, regeneration was noted, characterized by numerous brownish-black cells that displayed unique features in contrast to the adjacent mature cells (Figure 1B). Regeneration was commonly found throughout the explants within the entire stress period, but near the edges, the density of the regeneration was higher (Figure 1C). However, after certain periods, cell damage started to increase under high-stress conditions (Figure 1D).

### 2.2. Functional Annotation of Stress-Related Genes and Pathways

Pathway enrichment study categorized 409,468 unigenes into 424 different pathways utilizing our transcriptome data. These pathways encompassed various domains, including metabolism, cellular processes, organismal systems, genetic information processing, and environmental information processing [45,46,47]. Our previous work [18] revealed significant overexpression in several pathways, including nitrogen metabolism, pyruvate metabolism, the citrate cycle (TCA cycle), the peroxisome pathway, the ribosome pathway, glycolysis/gluconeogenesis, and carbon fixation in photosynthetic organisms. Furthermore, this study predicted that under stressful circumstances, the TCA cycle and pathways linked to glycolysis were upregulated. Also, transcriptome analysis indicated that the glycolysis pathway has a significant concentration of differentially expressed genes (DEGs) (Figure 2A) involved in *U. prolifera*’s response to temperature changes (Appendix A). Numerous enzyme genes were also shown to be differentially accumulated in *U. prolifera* during high-temperature–light stress. These genes, which included *pgm*, *GCK*, *G6PC*, *GALM*, *G6PE*, *GPI*, *FBP*, *PFK9*, *ALDO*, *TPI*, *gapN*, *PGK*, *PGAM*, *MINPP1*, *ENO*, *PCK*, *pckA*, *PK*, *ppdK*, *PDC*, *DLD*, *ALDH*, and *ADH*, were crucial for glycolysis (Figure 2A). Under conditions of high temperature and light stress, the expression of several genes involved in glycolysis changed. *GCK*, *G6PC*, *G6PE*, *GPI*, *FBP*, *TPI*, *PGK*, *PGAM*, *MINPP1*, *PCK*, *pckA*, *PK*, *PDC*, and *ADH* were upregulated, whereas *GALM*, *ENO*, and *DLD* were downregulated. In addition to *PFK*, *pgm*, *gapN*, *ALDO*, *ppdK*, and *ALDH*, there was also downregulation (Figure 2B).

### 2.3. DNA Methylation Characteristics of Ulva prolifera

Whole-genome bisulfite sequencing (WGBS) provided data for both global and site-specific DNA 5mC methylation, encompassing sequencing metrics and methylation calls. The cytosine content was reported as an average of 871,998.5. The methylation fraction for each sample was evaluated in each context to illustrate overall methylation levels. Methylation ratios were determined by dividing the count of methylated reads from bisulfite-converted sequences by the total number of reads at that specific location. Our study revealed that, on average, methylation was seen in roughly 72% of CpG contexts, about 10% of CHG contexts, and approximately 3% of CHH contexts (Figure 3A). From a genome-wide viewpoint, the *U. prolifera* genome exhibited around 1.18% cytosine methylation (Figure 3A). This was ascertained by calculating the ratio of methylation sites to total genome size, normalized according to the number of identified sites from the sequencing run (CG + CHH + CHG/CG). The methylation was distributed throughout all cytosine contexts, predominantly occurring at CpG sites.

Changes in global and site-specific DNA methylation of *U. prolifera* between the control group and the treatment group were determined from the WGBS data. Global CpG DNA methylation decreased (hypomethylation). No significant changes in global DNA methylation were found in the CHG and CHH contexts (Figure 3B).

### 2.4. The Measures of H_2_O_2_, Catalase Activity, Peroxidase Activity, and Pigment Composition Levels

The maximum chlorophyll-a level was observed at 12 h, significantly exceeding that of previous sampling points (*p* < 0.05). On the other hand, Chlorophyll-b (chl-b) levels showed approximately similar levels at 12, 24, and 36 h later, significantly decreased (*p* < 0.05) compared to other sampling points. Carotenoids exhibited similar findings comparable to those of Chl-b (Figure 4A). With prolonged stress duration, the levels of H_2_O_2_ and peroxidase significantly increased (*p* < 0.05), whereas the content of catalase significantly decreased (*p* < 0.05) in comparison to the control group (Figure 4B–D).

### 2.5. Validation of RNA-Seq Results with RT-qPCR

For the purpose of validating the patterns of DEGs identified in the RNA-seq expression analysis, we carefully chose nine genes that are known to play a significant role in the glycolysis pathway (*GCK*, *G6PE*, *PFK10*, *FBP*, *ALDO*, *PGK*, *ENO*, *pckA*, *PK*). Through a comprehensive analysis of both RNA-seq data and RT-qPCR results, it was observed that the expression levels of these genes exhibited significant changes in response to stress, and the majority of the RT-qPCR results showed a significant correlation with the RNA-seq results (*p* < 0.05) (Figure 5), which demonstrated the excellent accuracy of the RNA-seq analysis.

## 3. Discussion

The present research illustrated that *U. prolifera* contains both monospores and archeospores from uniseriate and multiseriate thalli, which develop into asexual spores that are subsequently released via cell wall rupture (Figure 1) [48]. The procedure typically ensues within 3–4 days, enabling the detached thallus to withstand adverse environmental conditions [12]. These findings are consistent with prior research on callus formation provoked by injured macroalgae [12,49]. These observations suggested a potential adaptive mechanism that may serve to shield the parent plant from stressful environments [50].

Overall, these findings suggested the hypothesis that *U. prolifera* can maintain abiotic stress tolerance as a physiological response to heat and light stress via asexual reproduction and develop stress resilience through the processes related to stress memory [11,32,50,51]. Collectively, these findings highlighted the capacity to adapt and proliferate effectively under environmental stress, offering vital insights into its ecological success and ability to create massive blooms.

Moreover, our research indicated that when macroalgae experience environmental stress, they respond by upregulating genes associated with energy production [18,31,46]. The prior study demonstrated an improvement in carbohydrate metabolism pathways and the activation of other pathways, such as glycolysis and lipid metabolism [18]. These variations facilitated essential energy for crucial survival processes through genetic regulation and regeneration [18,35,46,52]. Analysis of our previous transcriptome data revealed that differentially expressed genes (DEGs) were primarily concentrated in the glycolysis pathway (Figure 2A).

Moreover, our research indicated that when macroalgae experience environmental stress, they respond by upregulating or downregulating genes associated with energy production [18,31,46]. The prior study demonstrated an improvement in carbohydrate metabolism pathways and the activation of other pathways, such as glycolysis and lipid metabolism, which facilitated essential energy for crucial survival processes through genetic regulation and regeneration [18,35,46,52].

Analysis of our previous transcriptome data revealed that differentially expressed genes (DEGs) were primarily concentrated in the glycolysis pathway (Figure 2A). Key DEGs such as hexokinase and 6-phosphofructokinase-1 played a crucial role in these processes (Figure 2B) [53,54]. A notable increase in 3-PGA was observed, potentially replacing glucose in downstream glycolysis to generate pyruvate. This switch altered influenced the upstream pathways, leading to modifications in the gene expression of hexokinase and 6-phosphofructokinase, as well as an increase in pyruvate kinase (EC: 2.7.1.40). Additionally, Figure 2B demonstrated elevated level of fructose-6-phosphate, indicating the activation of the glycolysis/gluconeogenesis pathway as an energy supply mechanism under high-temperature–light stress [44,54,55,56]. 

Elevated temperature–light stress promoted glycolysis and electron transfer, enabling *U. prolifera* to generate ATP and other essential metabolites by modulating transcript levels of the glycolysis/gluconeogenesis pathway. Pyruvate generated through the glycolysis pathway enters the mitochondria, resulting in the synthesis of ATP and metabolites through the tricarboxylic acid (TCA) cycle. Key TCA cycle genes involved in this process are isocitrate dehydrogenase, 2-oxoglutarate dehydrogenase, and dihydrolipoamide succinyltransferase [57]. However, irregularities in TCA enzyme activity under abiotic stress may provoke aerobic glycolysis, whereas pyruvate dehydrogenase (PDH) facilitates mitochondrial entry of pyruvate, while PDH kinases impede it [58]. Notably, DNA methylation played a significant role in this step. We have performed DNA methylation analysis to further investigate methylation levels under high-temperature and light-intensity stress.

In bacteria, plants, and eukaryotic algae, DNA methylation is an essential epigenetic alteration that significantly affects gene regulation, genomic stability, and adaptability [56]. DNA methylation is used by plants not only to silence transposable elements but also to control gene expression during development and in response to environmental stimuli. In contrast, bacteria usually use methylation for defense through restriction–modification mechanisms. Likewise, differential DNA methylation has been connected to the control of pathogenicity-related genes in the filamentous plant pathogen *Verticillium dahlia* [59].

Our research has shown that DNA methylation in the green macroalga *U. prolifera* acts as a positive regulator of stress responses, suggesting an epigenetic mechanism that is flexible and suitable for dynamic marine environments [60]. Notably, DNA methylation in *U. prolifera* was predominantly seen in the CG context, including around 71.7% of all methylation occurrences. The methylation of CHG and CHH constitutes 10.2% and 2.91% of the total, respectively. This distribution resembled that observed in a different green alga, *Scenedesmus acutus*, where methylation predominantly occured at CG sites [61,62]. Their role in seaweeds is more intricate than in humans, as well as higher plants, where CHG and CHH methylation often regulate gene silencing and heterochromatin formation. These non-CG methylation patterns may contribute to the preservation of genomic integrity and the regulation of epigenetic plasticity in algae in response to environmental challenges. Methylation of CHG and CHH specifically may stabilize repetitive sequences and inhibit transposable elements, hence enhancing adaptability and stress resilience. In *Apostichopus japonicus* and other marine organisms, CHH methylation has been associated with heightened epigenetic responsiveness to abiotic stress [62].

Overall, this study focused on a coherent relationship between DNA methylation and glycolysis-related genes in *U. prolifera* under high-temperature–light stress and tried to demonstrate how changing methylation levels harmonized with significant upregulation or downregulation of key glycolytic genes validated by RT-qPCR (Appendix A) [63]. This correlation suggested that DNA methylation may function as a positive regulator of stress-responsive gene expression in *U. prolifera*. Although this study established a correlation basis for abiotic stress resistance mechanisms by utilizing epigenetic and transcriptomic data, there are still several gaps. Future studies are necessary, focusing on differentially methylated regions overlapping differentially expressed genes to optimize the abiotic stress conditions for survival and growth. Also, in this study, only two biological replicates were used, which may limit statistical power. Future studies with more replications will help to strengthen the findings.

## 4. Materials and Methods

### 4.1. Algal Strain Collection and Culture Conditions

On 16 December 2022, *U. prolifera* samples were gathered at Garorim Bay, which is located on South Korea’s west coast (122°03′ E, 39°07′ N). As soon as the sample had been collected, it was stored on dry ice and brought to the lab in less than three hours. The samples were cleaned in the lab using ddH_2_O to rid of stains, planktonic microbes, and other algae. After that, the sample was subsequently acclimated to filtered seawater with a temperature of 18 °C, a pH of 7.9, a salinity of 32, and a light intensity of 100 µmol photons m^−2^ s^−1^. The day/night cycle throughout the acclimatization period was 12–12 h.

### 4.2. Exposure Experiment and Sample Collection

Two treatment groups of *U. prolifera* (3 g) were created: high stress (H) and control (C) groups. Each treatment group has two replications. The algae were divided into treatment groups using sterile equipment. Each replicate was grown independently in conical flasks with 200 mL of culture media, and different treatment conditions were applied to each flask. For the stress group (H), elevated temperatures and light intensity stress (30 °C and 300 µmol photons m^−2^ s^−1^) were applied. We selected the experimental temperature and light based on previously published data [64,65] and pre-experiments that we conducted in our laboratory. The temperature of the control group (C) was kept at 20 °C, and light intensity was 100 µmol photons m^−2^ s^−1^. There were two duplicates in each group, and there was two grams of algae in each conical flask. Following stress exposure, 0.1 g of algae was taken from each group at 48 h, frozen in liquid nitrogen, and stored at −80 °C for further experiment. Collection of algal strains and their respective culture conditions.

### 4.3. Morphology Observation and Image Processing

Following a 48 h acclimatization to high-temperature–light stress conditions, the regeneration potential of Ulva was evaluated using morphological observation. Thalli were captured using an Olympus light microscope connected with a camera. The quantification of regeneration involved measuring the number of branches per centimeter along the major axis with ImageJ software (version 1.41). The process involved opening picture files of *U. prolifera* in ImageJ and defining the spatial scale using the Analyze/Set Scale dialogue. The segmented line tool was utilized to delineate the outlines of the principal axis and its branches. The analyze function was then utilized to ascertain the length of each structure. To calculate the branch counts per centimeter of the primary axis, the length of the primary axis was initially measured, the branches originating from the primary axis were counted manually, and the total branch count was divided by the axis length. To reduce operational mistakes, each measurement was conducted thrice, and the average value was utilized as the representative outcome. This method enabled a precise assessment of *U. prolifera*’s regeneration capacity under environmental stress caused by fluctuations in temperature and light.

### 4.4. Transcriptomic Data Analysis

We utilized our previously released transcriptome data to compare the findings (Jahan et al., 2024 [18]). Paired-end sequencing utilizing Illumina technology was employed to analyse the libraries and discover transcripts associated with diverse biological activities. The RNA sequence files have been submitted to the NCBI under accession numbers PRJNA1061775 and PRJNA1061781. Prior to performing a differential gene expression analysis, the read counts for each sequenced library were adjusted utilizing the edgeR (v4) software program and a singular normalization scaling factor. The density distribution of expression levels, quantified by log2 (CPM+2), in each library of differentially expressed genes (DEGs) demonstrated that the expression levels were analogous in the H48 and C libraries. We employed a relational model to compare the transcriptome libraries of H to C, which served as the control group. The Gene Ontology database at http://www.geneontology.org/ (accessed on 24 October 2023) has associated GO keywords with each candidate gene. The GO project categorizes genes into three groups: biological process, cellular component, and molecular function, based on their annotations. Subsequent to annotation, we conducted a pathway-based analysis using the Kyoto Encyclopaedia of Genes and Genomes (KEGG) (http://www.genome.jp/kegg/ko.html: accessed on 24 October 2023) to find the metabolic pathways that have high enrichment for the candidate genes. A biologist employed a methodology to discover genes with differential expression. Statistical analysis was performed utilizing the exact test with edgeR and the Fold Change method for each comparison pair. Significant findings were selected based on particular criteria, using the raw *p*-value from the exact test and the absolute value of |fc|.

### 4.5. Whole Genome Bisulfite Sequencing Analysis

Genomic DNA was extracted using the AccuPrep Plant Genomic DNA Extraction Kit (K-3031, Bioneer Corporation, Daejeon, Republic of Korea), a modified, combined method to isolate genomic DNA in accordance with the manufacturer’s instructions. The Qubit BR test was used to detect the amount of DNA, and 1% agarose gel electrophoresis was used to confirm the presence of high molecular weight DNA (10 kb). Ultrasound was used to randomly fragment the DNA to 200–300 bp. End repair was performed with 3’plus A base and a joint. DNA is treated with bisulfite, methylated cytosine stays the same, whereas unmethylated cytosine is converted into uracil.

The BSMAP: whole genome bisulfite sequence MAPping tool version 2.90 (https://code.google.com/archive/p/bsmap/: 20 January 2024) pipeline from Zymo Research (Irvine, CA, USA) was used to process and analyze *U. prolifera* samples. Analysis was performed utilising biological replicates from both the control and temperature stress treatment groups. In accordance with the manufacturer’s guidelines, DNA libraries were constructed with Accel Methyl-Seq DNA libraries and 100 ng of fragmented DNA as input. To put it briefly, bisulfite-converted DNA is first denatured for two minutes at 95 °C before being promptly cooled on ice. After that, the pre-made Adaptase Reaction Mix is added, and the mixture is then incubated in a thermocycler. The Extension Reaction Mix is then added, incubated, and then cleaned out with ethanol and beads. Once the Ligation Reaction Mix is ready, the samples are ligated at 25 °C, followed by another clean-up. Amplification and cleanup come next, after which the Indexing PCR Reaction Mix and indexed adaptor primers are added. An Illumina NovaSeq6000 sequencer (Illumina Inc., San Diego, CA, USA) was used to perform 100 bp paired-end sequencing. During the trimming process, Trim Galore version 0.6.10 (https://www.bioinformatics.babraham.ac.uk/projects/trim_galore/: 13 March 2024) and FastQC version 0.11.7 (http://www.bioinformatics.babraham.ac.uk/projects/fastqc/: 13 March 2024) produced quality control for the FASTQ files.

The raw sequence reads are filtered according to quality after sequencing. Additionally, the raw sequence reads are stripped of the adaptor sequences. BSMAP, which is based on the SOAP (Short Oligo Alignment Program), maps the trimmed reads to the reference genome. Only reads that are uniquely mapped are chosen for sorting, indexing, and PCR duplicate removal using SAMBAMBA (v0.5.9). The ’methylatio.py’ script in BSMAP is used to extract the methylation ratio of each and every cytosine position from the mapping data. The “# of C/effective CT counts” for each cytosine in CpG, CHH, and CHG were the findings of the coverage profiles. The table browser feature of the UCSC genome browser is used to annotate each cytosine locus in CpG, CHH, and CHG. Each gene’s functional location (promoter regions, which are defined as exons, introns, and upstream 2 kb of the transcription start site) is included in the annotation, along with the transcript ID, gene ID, strand, and CpG island.

### 4.6. Quantification of Methylation Level and Calling

#### 4.6.1. Mapping to Reference Genome

Based on the SOAP (Short Oligo Alignment Program), BSMAP was used to align the cleaned reads to the other. The statistics derived from the SOAP algorithm are displayed in Table 1. In methylation calling, the quantity of uniquely mapped reads, non-unique mapped reads, deduplicated reads, and analyzed reads can be checked.

#### 4.6.2. Alignment QC

Qualimap 2.2 was employed to evaluate the quality of the alignment data (a BAM file). A series of informative graphs is generated, accompanied by a summary of the alignment’s essential statistics (ACGT content, mean and standard coverage by chromosome, insert distribution, etc.).

#### 4.6.3. Methylation Level Calling

Methyratio.py in BSMAP is used to calculate the methylation ratio of each and every cytosine that satisfies a CT count greater than 1. Methylation is referred to by a single read for the areas that are covered by both ends of a read pair. For each of the three sequence contexts (CG, CHG, and CHH), the coverage profile findings are summed up as the number of C/effective CT counts. The methylation coverage for each sample is shown in Table 2 below.

### 4.7. The Measures of Catalase Activity, Peroxidase, and Hydrogen Peroxide Levels

The catalase activity, peroxidase, and hydrogen peroxide levels were assessed following the established methodology (DoGenBio, Seoul, Republic of Korea). In summary, 0.1 g of algal tissue was obtained and homogenized with 1x assay buffer in an ice bath. A tenfold dilution was performed using the identical assay buffer. The mixture underwent centrifugation at 12,000 rpm for 10 min at 4 °C. The activities of peroxidase, CAT, and H_2_O_2_ were quantified utilizing commercial assay kits. The absorbance was recorded at 570 nm for peroxidase and at 560 nm for both CAT and H_2_O_2_.

### 4.8. Pigment Composition

Chlorophyll a (chl a), chlorophyll b (chl b), and carotenoids from *U. prolifera* were isolated and quantified using the method described by Wellburn (1994) [66]. Approximately 0.02 g of fresh weight of *U. prolifera* thalli was immersed in 5 mL of 100% methanol and maintained at 4 °C in the dark for 24 h [67]. The absorbance was quantified via a spectrophotometer (Orion AquaMate 8000, Thermo Fisher Scientific Solutions LLC, Seoul, Republic of Korea) at wavelengths of 470, 653, and 666 nm, respectively. The pigment concentrations were reported as mg g^−1^ fresh weight.

### 4.9. Functional Validation

In order to validate the findings from Illumina sequencing, a selection of 9 differentially expressed genes from the high stress group (*GCK*, *G6PE*, *FBP*, *PFK10*, *ALDO*, *PGK*, *ENO*, *pckA*, and *PK*) was chosen depending on their function for RT-qPCR analysis. All genes are related to the glycolysis pathway. For each gene, we used three biological replications. Primer5 software (Premier Biosoft International, Palo Alto, CA, USA) was utilized to design the primer (Appendix A). An internal control was used in the RT-qPCR analysis, specifically the β-actin gene. cDNA was synthesized using Prime ScriptTM RT reagent Kit, Takara (Takara, Tokyo, Japan), following the manufacturer’s protocol. The RT-qPCR was performed using SYBR Premix Ex Taq II from Takara (Takara, Tokyo, Japan) following our previous study [18]. The 2^−ΔΔCT^ method was utilized to analyze the data. One-way analysis of variance was conducted to compare the effect of different stresses on gene expression in *U. prolifera* tissue. This was followed by Tukey’s multiple comparison tests to further examine the results. A *p*-value below 0.05 was considered statistically significant.

## 5. Conclusions

Our findings indicated that DNA methylation regulates gene expression and genomic stability in *U. prolifera* under stress conditions. Whole-genome bisulfite sequencing demonstrated significant methylation alterations in response to high-temperature–light stress, emphasizing CG enrichment. We identified an inverse correlation between CG methylation levels and gene expression, particularly in the promoter and gene body regions of stress-responsive genes, utilizing transcriptome profiling and methylome data.

The methylation patterns of CHG and CHH were more diverse and concentrated in transposable elements and intergenic regions. In reaction to stress, non-CG methylation types exhibited notable dynamism, indicating they may limit transposon activity and safeguard genomic integrity amid environmental fluctuations. CG methylation modulates gene expression, whereas CHG/CHH methylation fortifies the genome, demonstrating an adaptive methylation landscape that enables *U. prolifera* to precisely adjust gene expression and preserve epigenomic integrity under stress.

Understanding epigenetic regulation can help introduce new *Ulva* strains that are resistant to heat and light stress, helping steady biomass production in climate change conditions. These findings have important implications for managing aquaculture production, hence expediting the domestication of *U. prolifera* for industrial use and regulation or mitigation of algal blooms. Our study established a basis for utilizing epigenetic insights in the ecological management and sustainable use of bloom-forming green macroalgae.

## Figures and Tables

**Figure 1 ijms-26-06169-f001:**
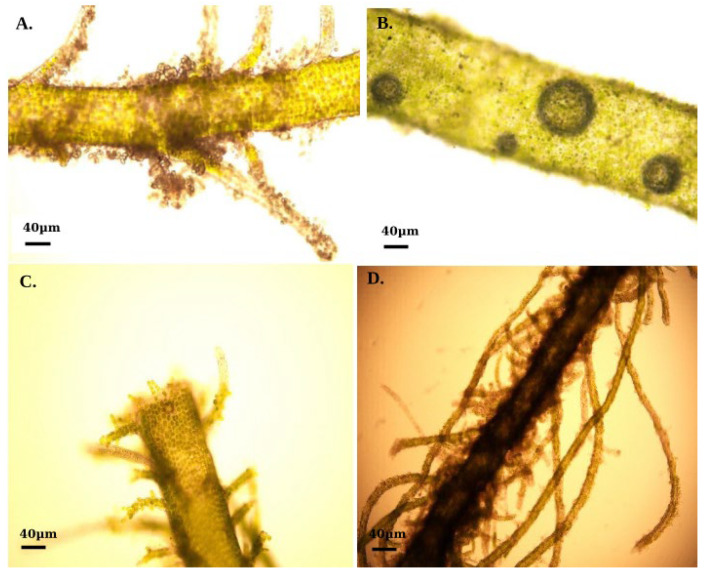
Morphological characteristics of the regeneration of *Ulva prolifera* under a microscope. (**A**) Rapid regeneration within 24 h of stress. (**B**) Numerous brownish-black cells with unique features appeared. (**C**) Higher regeneration near the edges. (**D**) Cell damage condition.

**Figure 2 ijms-26-06169-f002:**
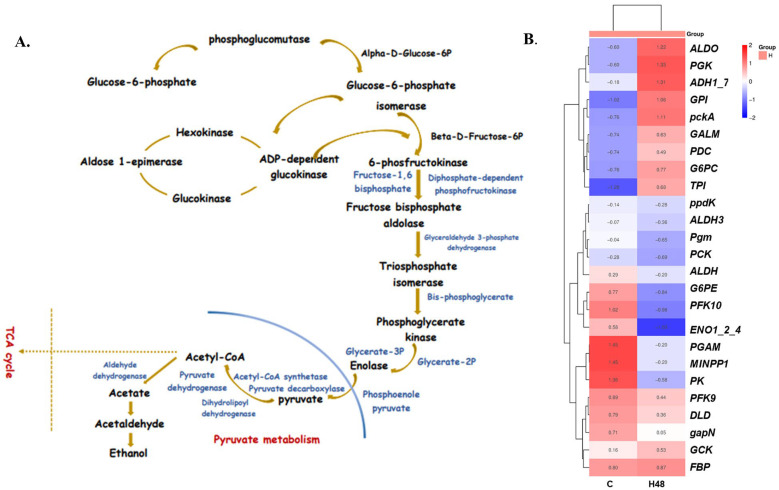
(**A**) Glycolysis pathway. (**B**) Significantly up-regulated or down-regulated genes at the Glycolysis pathway; H48—High stress group sample at 48 h; C—Control group; H—High stress group. Blue words indicated enzyme; black words indicated gene.

**Figure 3 ijms-26-06169-f003:**
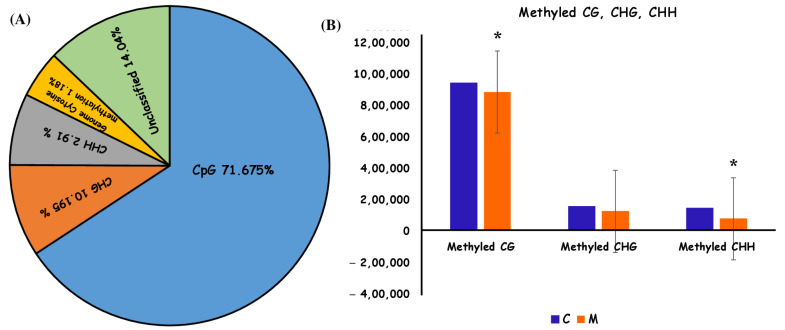
Genomic DNA methylation characteristics of *Ulva prolifera.* (**A**) Graphical representation of genomic methylation in different contexts: CpG (71.675%), CHG (10.195%), CHH (2.961%), and genome cytosine methylation (1.18%). (**B**) The distribution of 5mC DNA methylation CpG, CHG, and CHH cytosine contexts. “*” represents significant differences among samples (*p* < 0.05).

**Figure 4 ijms-26-06169-f004:**
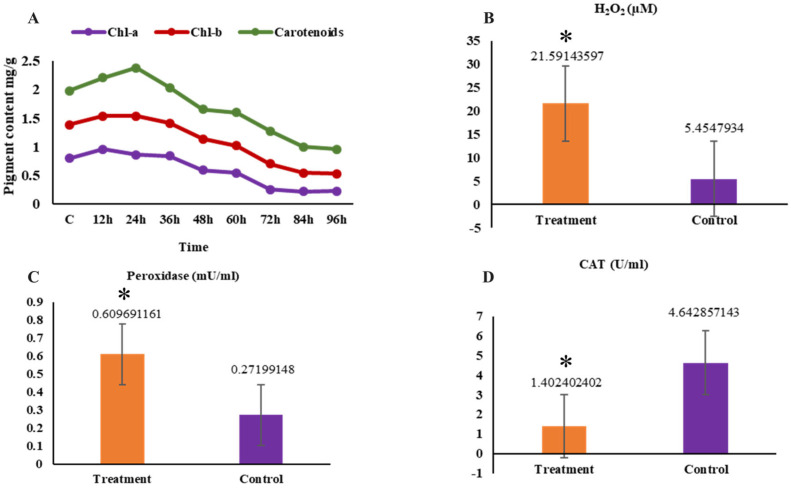
(**A**). Pigment content under stressful conditions. (**B**) Antioxidant activity between the treatment and control groups (H_2_O_2_). (**C**) Antioxidant activity between the treatment and control group (Peroxidase). (**D**) Antioxidant activity between the treatment and control group (CAT). “*” represents significant differences among samples (*p* < 0.05).

**Figure 5 ijms-26-06169-f005:**
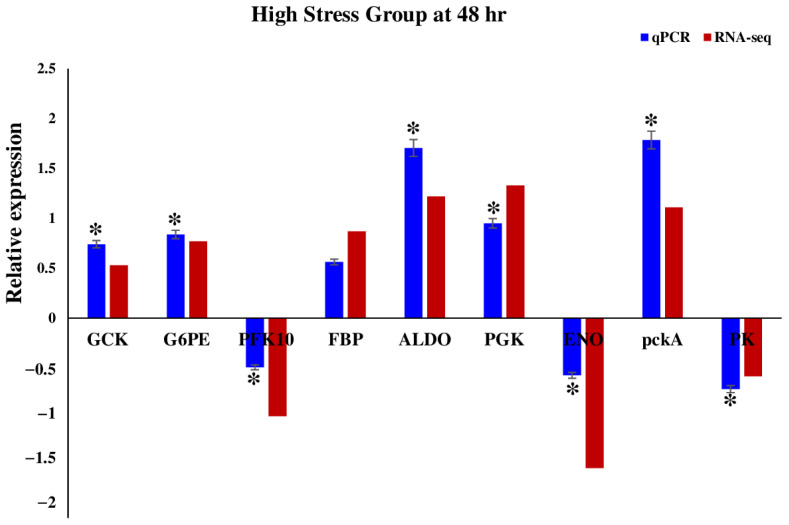
The relative mRNA expression pattern of different DEGs from the glycolysis pathway under HighStress Group at 48 h (Blue bar indicates qPCR results, red bar indicates RNA-seq results, “*” represents significant differences (*p* < 0.05)).

**Table 1 ijms-26-06169-t001:** Mapping data stats.

Sample ID	# of Trimmed Read Bases (bp)	# of Uniquely Mapped Reads	Deduplicated Reads (Deduplicated by SAMBAMBA Tools, % Out of Mapped Reads)	Analyzed Reads in BSMAP Methylation Calling
C1	10,590,317,370	184,798	166,728	166,728
C2	10,014,659,627	193,368	174,162	174,162
M1	10,032,087,532	190,324	170,528	170,528
M2	9,700,697,089	198,094	179,242	179,242

**Table 2 ijms-26-06169-t002:** Methylated coverage in CpG, CHG, and CHH.

Sample ID	Total CG	Methylated CG	Methylated CG (%)	Total CHG	Methylated CHG	Methylated CHG (%)	Total CHH	Methylated CHH	Methylated CHH (%)
C1	1,156,390	836,169	72.31%	1,077,017	143,952	13.37%	2,313,628	147,960	6.40%
C2	1,241,700	908,617	73.18%	1,162,330	146,547	12.61%	2,446,691	121,649	4.97%
M1	1,170,302	846,696	71.35%	1,144,518	125,820	10.99%	2,333,659	74,316	3.18%
M2	1,246,328	897,301	72.00%	1,218,353	114,527	9.40%	2,472,124	65,163	2.64%

## Data Availability

The original transcriptomic data presented in the study are available in the NCBI Sequence Read Archive (SRA) database with the accession numbers PRJNA1061775 and PRJNA1061781.

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
