# Peer review of "Genome-Wide DNA Methylation and Transcription Analysis Reveal the Potential Epigenetic Mechanism of Heat–Light Stress Response in the Green Macro Algae Ulva prolifera"

_ijms, 2025, doi:10.3390/ijms26136169_

Round 1
Reviewer 1 Report
Comments and Suggestions for Authors
The results of the article fill the gap in the field of environmental epigenetics for non model organisms by combining DNA methylation with the ecological adaptability of the green macro algae Ulva prolifera, indicated that epigenetic modifications may mediate long-term adaptive evolution,
- Not clearly stated how to prove that methylation directly leads to gene regulation
- 1, Scale bar must be here in each picture.
- 2, authors should give sharper picture.
- Legend in Fig 3A should be clearer.
- Fig 4, Figure Legend “H2O2”should be H2O2
- in Fig 5, how many biological repeats did you do? How to calculate the significant difference? What is the “*”represent?
Author Response
Thank you very much for taking the time to review this manuscript. Please find the detailed responses below and the corresponding revisions/corrections highlighted/in track changes in the re-submitted files
The results of the article fill the gap in the field of environmental epigenetics for non-model organisms by combining DNA methylation with the ecological adaptability of the green macro algae Ulva prolifera, indicated that epigenetic modifications may mediate long-term adaptive evolution,
Comment 1: Not clearly stated how to prove that methylation directly leads to gene regulation
Response 1: Thank you for your valuable suggestion. While the annotation file for Ulva prolifera genome has not been available yet which limits showing direct relationship between methylation and gene. To address the reviewer’s concern, we used stress-responsive genes involved in high temperature-light intensity stress resistance and differentially methylated regions (DMRs) using our bisulfite sequencing data to demonstrate the epigenetic regulation of key genes under high temperature-light intensity stress (Figure S1, Line 287).
Comment 2: 1, Scale bar must be here in each picture.
Response 2: Thank you for pointing out this. We agree with this comment. Therefore, we have revised the figure and included Scale bar. (Figure 1, Line 116)
Comment 3: authors should give sharper picture.
Response 3: Thank you for pointing out this. We agree with this comment. Therefore, we have revised all the figures and included sharper picture.
Comment 4: Legend in Fig 3A should be clearer.
Response 4: Thank you for pointing out this. We agree with this comment. Therefore, we have tried to make the legend in Fig 3A clearer (Line 163-167).
Comment 5: Fig 4, Figure Legend “H2O2” should be H2O2
Response 5: Thank you for pointing out this. We agree with this comment. Therefore, we have changed “H2O2” to H2O2 (Line 185).
Comment 6: in Fig 5, how many biological repeats did you do? How to calculate the significant difference? What is the “*”represent?
Response 6: Thank you for your comment. In Fig. 5 we have done three biological repeats and one-way analysis of variance followed by Tukey's multiple comparison tests was carried out to calculate the significant difference. P-value below 0.05 was considered statistically significant (Line 445; 450-453).
“*” represent significant difference among sample) (P < 0.05) (Line 201-202)
Reviewer 2 Report
Comments and Suggestions for Authors
Please find my suggestions in the attached file.

The English writing should be edited by native English speakers.
Author Response
Thank you very much for taking the time to review this manuscript. Please find the detailed responses below and the corresponding revisions/corrections highlighted/in track changes in the re-submitted files
This manuscript presents a multi-level investigation into the stress adaptation mechanisms of Ulva prolifera under high temperature and light conditions, integrating physiological measurements, transcriptomic profiling, and whole-genome bisulfite sequencing. The research question is timely, and the approach is comprehensive. With revision for language, clarification of experimental design, and improved data visualization, this manuscript would make a strong contribution to the literature on algal stress response and epigenetics.
General Suggestions and Comments
Comment 1: Experimental Design and Replication
- Only two biological replicates per group (n=2) are reported (line 287-288), which limits statistical power. This limitation should be explicitly acknowledged and discussed.
Response 1(a): Thank you for pointing out this. We agree with this comment. Therefore, we have acknowledged and discussed this limitation in our manuscript (Line-293-295).
- Clarify whether the RNA-seq and WGBS samples came from the same biological material and time points. If RNA-seq and WGBS were performed on samples collected at different time points or from different biological sources, the correlation between DNA methylation and gene expression would lack direct comparability, thereby weakening the validity of any inferred regulatory relationships.
Response 1(b): Thank you for pointing out this. In our experiment RNA-seq and WGBS samples came from the same biological material. There was some misinformation in the manuscript. We have revised this portion accordingly (Figure 2, Line 143; Line 19-192)
- Figures and Data Presentation
- Figure 2A/B (Glycolysis Pathway) lacks sufficient annotation. Please include (is the process of adding explanatory notes, comments, or labels to a text, image, video, or dataset to provide additional information, clarification, or context. It serves different purposes depending on the field)
enzyme/gene names and indicate up/down-regulation clearly.
Response 2 (a): Thank you for pointing out this. We agree with this comment. Therefore, we have provided sufficient annotation for Figure 2A/B (Glycolysis Pathway) (Line 144-147, Table S1))
- The methylation results should be supported by genome browser visualizations or specific locus examples. To strengthen the interpretation of the methylation data, locus-specific examples showing DNA methylation differences at key stress responsive genes, along with corresponding expression changes.
Response 2 (b): Thank you for your valuable suggestion. While genome browser visualizations or specific locus examples are best to support methylation results, but the annotation for Ulva prolifera genome has not been available yet which limits the reliability of genome browser-based track alignment and visualization. To address the reviewer’s concern, we used stress-responsive genes involved in high temperature-light intensity stress resistance and differentially methylated regions (DMRs) using our bisulfite sequencing data to demonstrate the epigenetic regulation of key genes under high temperature-light intensity stress (Figure S1, Line 284-289).
- The discussion repeats several concepts, especially regarding glycolysis, TCA cycle, and stress response genes. Consider consolidating overlapping paragraphs to improve flow and conciseness.
Response 3: Thank you for pointing out this. We appreciate your suggestion. Therefore, we have revised the discussion section accordingly (Line 221-259)
- Although methylation and gene expression are described separately, integrated analysis (e.g., differentially methylated regions overlapping differentially expressed genes) is lacking. If such data exist, it should be included or at least mentioned as a limitation.
Response 4: Thank you for your suggestion. Though genomes for U. prolifera have been available, the annotation of this genome has not been published yet. As a results we could not integrated differentially methylated regions overlapping differentially expressed genes. We mentioned this as a limitation in our manuscript (Line 289-293).
Minor Suggestions
- Clarify the central conclusion in the Abstract more directly.
Response 1: Thank you for your suggestion. We appreciate your suggestion. Therefore, we have clarified the central conclusion in the Abstract more directly (Line 13-33).
- Some citations in the Introduction are outdated or redundant.
Response 2: Thank you for your suggestion. We appreciate your suggestion. Therefore, we have updated the citations (Line 43, 59, 59, 74, 86, 87, 88)
- Consider summarizing the key up/down-regulated genes in a table.
Response 3: Thank you for your suggestion. We appreciate your suggestion. Therefore, we have summarized the key up/down-regulated genes in table S1 (Line 134)
- Add specific implications for aquaculture, green tide management, or algal breeding in the Conclusion
Response 4: Thank you for your suggestion. We appreciate your suggestion. Therefore, we have added specific implications for aquaculture, green tide management, or algal breeding in the conclusion (Line 468-474)